# Application of Grafting Method in Resistance Identification of Sweet Potato Virus Disease and Resistance Evaluation of Elite Sweet Potato [*Ipomoea batatas* (L.) Lam] Varieties

**DOI:** 10.3390/plants12040957

**Published:** 2023-02-20

**Authors:** Hong Huang, Haohao Han, Yayun Lei, Huanhuan Qiao, Daobin Tang, Yonghui Han, Zhenpeng Deng, Limin Mao, Xuli Wu, Kai Zhang, Jichun Wang, Changwen Lv

**Affiliations:** 1College of Agronomy and Biotechnology, Southwest University, Beibei, Chongqing 400715, China; 2Key Laboratory of Biology and Genetic Breeding for Tuber and Root Crops in Chongqing, Beibei, Chongqing 400715, China; 3State Cultivation Base of Crop Stress Biology for Southern Mountainous Land of Southwest University, Beibei, Chongqing 400715, China

**Keywords:** grafting, SPVD, symptomatic detection, NCM-ELISA, QRT-PCR

## Abstract

Sweet potato virus disease (SPVD) is one of the main virus diseases in sweet potato [*Ipomoea batatas* (L.) Lam] that seriously affects the yield of sweet potato. Therefore, the establishment of a simple, rapid and effective method to detect SPVD is of great significance for the early warning and prevention of this disease. In this study, the experiment was carried out in two years to compare the grafting method and side grafting method for three sweet potato varieties, and the optimal grafting method was selected. After grafting with seedlings infected with SPVD, the symptomatic diagnosis and serological detection were performed in 86 host varieties, and the differences in SPVD resistance were determined by fluorescence quantitative PCR (qRT-PCR) and nitrocellulose membrane enzyme-linked immunosorbent assay (NCM-ELISA). The results showed that the survival rate of grafting by insertion method was significantly higher than that by side grafting method, and the disease resistance of different varieties to sweet potato virus disease was tested. The detection method established in this study can provide theoretical basis for identification and screening of resistant sweet potato varieties.

## 1. Introduction

Sweet potato virus disease (SPVD) is a destructive disease of sweet potato caused by the coinfection of sweet potato feather mottle virus (SPFMV) and sweet potato chlorotic stunt virus (SPCSV). SPFMV is a member of the genus Potyvirus in the Potyviridae family, while SPCSV belongs to the genus Crinivirus in the Closteroviridae family [1]. Both viruses are single-stranded RNA (ssRNA) viruses [2,3,4]. Based on analysis of the genome sequences of virus isolates, three representative strains of SPFMV—the russet crack (RC), ordinary (O), and East Africa (EA) groups—have been recognized. Two distinct strain groups have been recognized for SPCSV—the East African (EA) and West African (WA) groups [2,4,5,6]. In 2002, Kreuze et al. (2002) determined the complete genome sequence of RNA1 and RNA2 of SPCSV, and the results showed that there were 9407 and 8233 nucleotides, respectively [7]. The infected plants show dwarfing, wrinkling, chlorosis, narrowing of leaves, bright veins of flowers and leaves [8]. It will cause 80–90% of the output loss, even when not harvested [9]. Especially for the asexual propagation plants such as lily, saffron [10,11], strawberry, sweet potato and chrysanthemum [12,13], the virus accumulates continuously in the plant during the cultivation periods, resulting in the degradation of varieties, quality decline, and seriously affecting the economic benefits.

In the middle of 1990, the occurrence of SPVD was reported in the United States, Argentina, Brazil, Peru and other American countries. In Asia, SPVD was also discovered in Israel [14]. In recent years, it has been reported that SPVD occurs frequently in China [15,16]. The occurrence and spread of SPVD can cause huge economic losses, and effective control measures have been rarely reported; thus, using SPVD resistance varieties was the most effective way to reduce the loss result from SPVD. Previous studies showed that SPVD-resistant varieties are often discovered in areas commonly infected with SPCSV, such as New Kawogo, which is one of the sweet potato varieties with the strongest SPCSV resistance [17]. However, there are few reports on SPVD resistance identification and screening of SPVD-resistant sweet potato varieties in China. Wang et al. (2014) used the method of artificial grafting virus scion in the field to identify the resistance and yield loss of 12 sweet potato varieties, and calculated the disease index of 12 varieties, providing reference for the resistance and susceptibility identification of sweet potato varieties. Lu et al. (2016) inoculated 10 varieties with SPVD using artificial friction virus, and determined that Yushu No.4 and Yushu No.12 showed high SPVD resistance, but 0841-14, Yushu 2, Xushu 22 and Ningzishu No.1 may be susceptible to SPVD [18].

Grafting is currently recognized as an effective technique to improve disease resistance, cold tolerance and high yield in crops [19,20,21]. Grafting method directly affects the formation of the callus and activity of cells at the grafting interface, as well as the growth of seedlings after grafting [22,23]. However, the common method of SPVD resistance identification is to use the infected Brazilian Petunia as scion for grafting inoculation. When being infected with SPFMV and SPCSV, the plants of Brazilian Petunia tend to be seriously dwarfed and the mortality rate is relatively high [24]. Therefore, Brazilian Petunia is not an ideal scion for SPVD resistance identification in sweet potato. It was determined that the survival rate of susceptible scions increased from 5% to 100% by using SPVD-infected sweet potato cultivar as scions and improved cleft-grafting technique [25]. Therefore, it is possible to identify the SPVD resistance of sweet potato varieties through grafting with disease seedlings using itself as stock.

At present, the methods for plant virus detection include symptomatic diagnosis, biological detection, electron microscope detection, NCM-ELISA, nucleic acid hybridization detection, rolling ring amplification, siRNA deep sequencing, PCR detection and so on [26]. ELISA and qRT-PCR are the most widely used methods for sweet potato virus detection [27].

In order to select the better grafting method used for SPVD resistance identification in sweet potato and to discover an efficient method for resistant varieties screening, in this study, the survival rates of two grafting methods were compared, and the better grafting method was applied to the SPVD resistance identification in elite sweet potato germplasm. Based on these results, an efficient method for SPVD resistance identification was established. The findings obtained will provide theoretical basis for identification and screening of virus-resistant sweet potato varieties.

## 2. Results

### 2.1. Comparison of Two Grafting Methods

The results obtained in both 2016 and 2017 showed that there was no significant difference in the survival rate between the two grafting methods in the early growth stage (Figure 1). During the 30-day period, the survival rate of plants using the side grafting method decreased greatly, with an average decrease rate of 38% in the two years, while the survival rate of plants using the insertion method decreased 17% (Figure 1), and the survival rate of the plants using side grafting method was lower than that of the plants using insertion method. Thirty days after grafting, the survival rate of the plants using insertion method was higher than that using side grafting method, indicating that the side grafting method was more harmful to the plants. Furthermore, in the plants of the three varieties, Ningzishu No.1, Xushu 22 and Yushu No.2, the survival rate of the plants using insertion method were all higher than that using insertion method, suggesting that the effect of insertion method is better than that of side grafting method, and insertion grafting method would be better than side grafting method in the identification of SPVD resistance.

### 2.2. Symptomatic Test and Cluster Analysis

After 15 days of SPVD inoculation in 2016, the disease indexes of tested plants were significantly different among sweet potato varieties, and varied between 0.71 and 35. Among the tested varieties, the disease index of Ning 4-6 was the highest, indicating that it is a SPVD-susceptible variety, while the disease index of Wanzi 51 was the lowest, indicating that it is a SPVD-resistant variety. Nanshu 95, Mianshu No.6 and Yushuwang also showed high disease indexes, indicating the three varieties were more susceptible to SPVD when compared with another tested varieties.

The disease indexes of other varieties were significantly lower than that of Ning 4-6. Thirty days after grafting (30 DAG), the disease indexes of most of tested varieties increased and were significantly higher than that of 15 days after grafting (15 DAG, Table 1). The disease index of Ningzishu No.1 was the highest, indicating this variety was SPVD-susceptible, while the disease index of Enshu No.2 was the lowest, only 2.86, which indicated Enshu No.2 was resistant to SPVD. From 15 DAG to 30 DAG, the disease index of 17 varieties such as Xuyushu 43 and Yushu H210 remained unchanged, while the disease indexes of other varieties increased, and the disease index of Ningzishu No.1 increased rapidly.

The cluster analysis of disease index was performed based on the disease indexes calculated in 15 DAG and 30 DAG in 2016 (Figure 2). In general, there were two varieties with disease index greater than 65, accounting for 3% of the tested varieties; there were 27 varieties with disease index between 28 and 65, accounting for 31%. There were 57 varieties with disease index between 0 and 27, accounting for 66%. These results indicate that there was a large proportion of SPVD-tolerant or resistant sweet potato varieties among the tested varieties.

Based on the results obtained in 2016, 12 SPVD-susceptible varieties and 11 SPVD-resistant varieties were identified repeatedly in 2017. The disease indexes of Wanzi 51, Enshu No.2 and Chaoshu No.1 were the lowest varieties among the tested varieties, and the disease indexes calculated in 2017 were only slightly increased when compared with that in 2016, indicating they might be varieties with high SPVD resistance. Ning 4-6, Ning Zishu 1 and Yushuwang showed the highest disease indexes and a large increase in disease indexes in 2017, indicating they were high-susceptible varieties (Table 2).

Twenty-three varieties tested in 2017 were divided into two subgroups based on disease indexes. The first subgroup included 12 varieties, whose SPVD disease index was greater than 27. The second subgroup included 11 varieties which did not show obvious symptoms, and the disease indexes were low (Figure 3). The classification result of disease index in 2017 is basically consistent with that obtained in 2016, indicating that the symptomatic detection method combined with cluster analysis could be used to identify and classify SPVD resistance of sweet potato varieties accurately.

### 2.3. NCM-ELISA Identification

The SPVD of eighty-six varieties were detected by NCM-ELISA in 2016. The results showed that the positive samples formed purple precipitates on the membrane, while the negative samples had no color reaction. Samples collected from 23 varieties were positive for SPFMV polyclonal antibody and 12 varieties were positive for SPCSV polyclonal antibody. Among them, there were 12 varieties positive for both SPFMV and SPCSV polyclonal antibody, indicating that these 12 varieties were infected both by SPCSV and SPFMV. Fifty varieties were simultaneously negative with the polyclonal antibodies of SPCSV and SPFMV, indicating that these varieties were not infected by SPVD. There were 21 varieties tested positive for SPFMV but negative for SPCSV, and these varieties also were not infected by SPVD (Table 3). In total, 17% of the tested varieties were revealed to have SPVD, and 83% were not infected by SPVD.

In 2017, the results of NCM-ELISA detection showed that 11 cultivars from 23 tested varieties were positive with polyclonal antibodies of both SPFMV and SPCSV, which proved that these varieties were infected by SPVD, and 21 varieties were tested positive on SPFMV but negative on SPCSV. The detection results of 23 varieties obtained in 2017 were consistent with those obtained in 2016 except for Yushu No.8, which was positive with polyclonal antibodies on SPFMV and SPCSV in 2016, positive on SPFMV but negative on SPCSV in 2017.

### 2.4. QRT-PCR Analysis

The expressions of genes of SPFMV and SPCSV could not be detected through qRT-PCR in leaves of all six varieties before grafting, indicating they were all healthy plants without SPVD. After grafting, in the leaves of Wanzi 51, Chaoshu No.1 and Enshu No.2, SPFMV and SPCSV could not be detected or the gene expressions were very low. In the leaves of Ningzishu No.1, Ning 4-6 and Yushuwang, high level of SPFMV and SPCSV were detected, indicating they were susceptible varieties, which was consistent with the results of symptomatic diagnosis and NCM-ELISA detection. These results showed that the SPVD detection and SPVD resistance identification results obtained using symptomatic diagnosis could be confirmed using qRT-PCR and NCM-ELISA methods with the accuracy of the results (Figure 4).

## 3. Discussion

Different grafting methods can affect the survival rate, and thus affect the symptom of the disease and the effect of disease resistance detection. The side grafting method needs to expose the wound of the scion seedling and the rootstock seedling. The rich mucin in the wound part makes it difficult to fix the interface, which affects the survival rate. On the other hand, the insertion method only exposes the wound of the scion seedling through wrapping and fixing together with the rootstock seedling, which causes the interface to recover more quickly and be less affected by the external conditions. In conclusion the grafting operation is relatively simple. Previous studies showed that the grafting method had the highest yield and the earliest fruit ripening, which was 23.65% higher than that of the control, and the harvest time was 7 days earlier than control. Compared with the attachment method and the control, the difference in yield reached a significant level [28]. Our results showed that the survival rate of the insertion method was significantly higher than that of the side grafting method, indicating that the grafting method could be used as a suitable method in the identification of SPVD resistance.

Rootstock, scion and sampling at different parts also affect the accuracy of virus detection. Ma et al. (2014) showed that the survival rate of grafted plants mainly depends on the compatibility of rootstocks, and is also affected by the characteristics of rootstocks themselves [29]. The rootstocks with strong stems have a higher survival rate than those with slender rootstocks. Wang et al. (2014) showed that the results of resistance identification were also related to grafted scion seedlings [30]. The actual virus accumulation in the scion may influence the symptom exhibition and resistance identification. In addition, previous study showed that the detection results of SPVD in different parts of sweet potato are also different, and the different parts of plants might show different sensitivity to the virus [31]. In order to improve the accuracy of resistance detection, other parts of sweet potato plant should be also detected.

The developmental stage and intensity of grafting also exhibit influence on virus detection [32]. The suitable period for grafting generally lasts from May to October. It was also shown that the disease index of sweet potato infected with SPFMV and SPCSV virus decreased with the postponement of grafting time [33]. Wang et al. (2014) used the method of artificial grafting virus scion in the field to identify the resistance of 12 sweet potato varieties, and the results showed that they were not resistant to SPVD [30]. However, Shangshu 19, Yushu No.8 and Zhengshu 20 were all resistant varieties in this experiment, and Lizixiang was a susceptible variety. These results were different from the results obtained in this study, probably because the environmental factors might influence the resistance of varieties, and different developmental stage of grafting were used in the two studies. Furthermore, in this study, NCM-ELISA and qRT-PCR detection were used to double-check the results of symptomatic detection. After natural infection and artificial friction tests on 10 sweet potato varieties, Lu (2016) determined that the SPVD resistance of Yushu 2, Xushu 22 and Ningzishu No.1 was relatively weak, while that of Ning4-6, Yushu 4 and Yushu 12 was relatively strong. In this study, Ningzishu No.1, Yushu 4 and Yushu 12 were identified as resistance varieties, which were basically consistent with the results of previous study [27].

The identification of SPVD resistance provides a strong guarantee for the breeding of high-quality sweet potato varieties. To ensure the accuracy of SPVD resistance identification, when grafting virus seedlings into sweet potato plants, it is necessary to not only improve the infection rate and survival rate, but also optimize the grafting method, stock selection, grafting technique and hygienic condition, time and intensity during the grafting period. It is necessary to ensure that sweet potato plants are infected by virus and minimize the influence of grafting methods. Our results showed that when compared with the side grafting method, the insertion method is more efficient and can be practically applied to sweet potato SPVD detection.

The results of NCM-ELISA showed that the varieties infected with SPCSV virus were often infected with SPFMV virus, and qRT-PCR showed that the total transcription level of SPCSV gene was 30 times higher than that of SPFMV (Figure 4). Previous study showed that the accumulation of SPFMV in sweet potato plants co-infected was 600 times higher than that in sweet potato plants infected alone, but there was no significant difference in SPCSV content [34]. Our results also showed that the accumulation of SPCSV in plants was higher than that of SPFMV, which was consistent with the previous studies.

There are two main pathways of virus transport in plants: the short-distance movement between adjacent cells through plasmodesmata and the long-distance transfer through vascular bundles [35]. The propagation modes of plant virus mainly include propagation of propagating materials, transmission of plants also through mechanical juice, transmission of media, etc. The virus will spread through propagating material. Plant viruses can infect healthy plants through mechanical wounds. Mechanical friction, pruning, and transplanting may cause plant viruses to spread through juice [36]. Natural infection and human friction are commonly used in virus inoculation or detection. Studies have shown that grafting can transmit SPVD more effectively than friction in sweet potatoes [37], possibly because SPCSV is restricted by phloem, and grafting would facilitate SPCSV transmission into the phloem. SPCSV could then enhance the multiplication of SPFMV and increase the titer of SPFMV in non-phloem tissues, causing severe SPVD symptoms [34].

In this study, the insertion method was applied to make the virus inoculation efficient and to exclude the influence of incomplete inoculation on SPVD anti-sensitivity identification. After grafting, symptom detection was applied to calculate the disease index of each variety, and cluster analysis was carried out based on the disease index, and the tested sweet potato varieties were divided into three subgroups. In addition, NCM-ELISA and qRT-PCR were combined to conduct SPVD resistance/susceptibility evaluation. The results obtained using NCM-ELISA and symptomatic test in two years were similar, and the SPCSV and SPFMV-infected varieties detected using NCM-ELISA detection showed severe symptoms. The SPCSV and SPFMV-uninfected varieties detected using NCM-ELISA were also resistant in symptomatic classification. However, the disease indexes calculated in Xinxiang and Yushu 8 were low, but SPFMV and SPCSV could be detected by NCM-ELISA in both of the two varieties. The possible reason might be because SPVD was accumulated in Xinxiang and Yushu 8, but did not display severe symptoms. In 2017, NCM-ELISA detection showed no SPVD in Yushu No.8, although its disease index was 35.71, and severe symptoms were exhibited. It might because the SPCSV content in the plants was very low and not easily detected by NCM-ELISA, but severe symptoms could also be exhibited.

It was shown that the NCM-ELISA results verified the symptomatic classification results, and qRT-PCR results were also consistent with those of cluster analysis, which further verified the accuracy of the SPVD resistance identification. Therefore, the SPVD resistance of sweet potato genotypes could be more accurately evaluated by a combination of insertion method, disease index, cluster analysis, NCM-ELISA and qRT-PCR.

## 4. Materials and Methods

### 4.1. Comparison of Two Grafting Methods

#### 4.1.1. Plant Materials

The purple-fleshed sweet potato variety Ningzishu No.1, white-fleshed variety Xushu 22 and Yushu No.2 were selected for comparative experiment of grating methods. Eighty-six elite sweet potato varieties were used for SPVD resistance identification.

#### 4.1.2. Experimental Design

For each variety, 300 healthy sweet potato seedlings, which were confirmed to be without virus infection by NCM-ELISA and qRT-PCR, were planted as grafted rootstocks, and 650 healthy Xushu 22 seedlings without virus infection were planted as grafted scion seedlings. The randomized blocks design was used for grafting method comparison with two replications. The 300 plants of each variety were planted in one plot and divided into three groups. A total of 100 untreated plants were used as the control group, 100 plants were grafted by side grafting and another 100 plants were grafted by insertion method. The entire experimental block was 38.4 m long and 16.0 m wide, and the passageway between plots was 0.8 m.

#### 4.1.3. Grafting

Two grafting methods, the insertion and side grafting methods, were compared in this study. Insertion grafting method (Figure 5A) consists of these steps: the single segment of seedling is used as scion, the leaves are removed and the seedling is cut into wedges, and an oblique opening is cut in the middle of stem below the cotyledon of rootstock sweet potato seedling, the wedged scion is inserted, and the grafting clip is placed at the grafting place after being bound with plastic grafting film for fixation [38].

Side grafting method (Figure 5B) is described by one previous report [30]: take the top of the sweet potato seedling of the scion, remove the leaves, cut out approximately 1 cm-long slope at the stem end, reach the cambium, and expose the oblique section; cut off the top of the healthy rootstock seedling, cut a slope on the stem with an area equivalent to that on the scion, stick the two cutting planes closely, then wrap them with plastic grafting film and fix them with grafting clip, and take the grafting clip after the grafting plants survives.

To guarantee that scions completely adhere to the subjected rootstocks, the plastic clamp was used to fix the place bounded with the plastic film (Figure 5).

#### 4.1.4. Grafting Survival Rate Calculation

Grafting survival rate was used to evaluate the effect of two grafting methods. The number of surviving seedings was recorded every five days. The grafting survival rate was calculated as follows:Grafting survival rate (%) = number of survival seedlings/number of grafted seedlings × 100.(1)

### 4.2. Identification of Resistance to SPVD in Sweet Potato

#### 4.2.1. Plant Materials

In 2016, 86 elite sweet potato varieties were used for SPVD resistance identification. Furthermore, in 2017, 23 susceptible and disease-resistant varieties were selected from the 86 varieties for experiments. The layout and planting of all the materials were the same as the previous three varieties.

#### 4.2.2. Virus Inoculation

The tested sweet potato varieties were infected with SPFMV and SPCSV virus by the grafting method. The top parts of the stems of sweet potato plants, which took on typical symptoms and were identified as SPVD-infected plants using NCM-ELISA and RT-qPCR, were used as scion. The seedling tips of healthy varieties were used as rootstock for grafting infection test. The rootstocks are just the plants from the same variety, and they were transplanted on 25 May 2016. Thirty days after transplanting, for each variety half of the plants was used as the rootstocks.

The healthy and diseased plants were determined using NCM-ELISA and qRT-PCR.

#### 4.2.3. Disease Index Calculation and Cluster Analysis

According to the disease condition of all plants at 15 and 30 DAG in 2016, their symptoms were observed, and the disease rate was calculated by the method described previously [30]. Disease index = Σ (number of sick plants at each grade × grade number)/(total plant number × 7) × 100.

SPVD disease grading standards were the following: Grade 0, no symptoms; Grade 1, mild symptoms, including slight chlorosis or purplish spots; Grade 3, moderate symptoms, including mellowing of green spots, pulse brightness, midrib mellowing, mottling and mosaic; Grade 5, the plants did not dwarf, but showed severe symptoms such as bright veins, mottling, greenish spots and mosaic; Grade 7, severe plant showed dwarfing, leaf deformities and severe fading. The disease index was analyzed with R studio.

#### 4.2.4. NCM-ELISA

At 30 DAG, three leaves of each variety were selected, and the leaf disc with a diameter of 1 cm was taken with a punch, mixed and ground and then placed on the nitrocellulose membrane, and then the SPFMV and SPCSV viruses were detected by NCM-ELISA detection kit (International Potato Center, CIP, Lima, Peru). The detection method was carried out according to the manufacturer’s instructions.

#### 4.2.5. RT-qPCR Detection

Two to three leaves were taken from each variety and the total RNA was extracted using the plant total RNA extraction kit (Tiangen Biochemical Technology Co., Ltd., Beijing, China), and the genomic DNA in the total RNA was removed by DNase I (Tiangen Biochemical Technology Co., Ltd., Beijing, China). A total of 1 μg of RNA was used as template. The first strand of cDNA was synthesized using the Prime Script RT reagent Kit (Ta Ka Ra, Dalian, China).

The con-infection of SPFMV and SPCSV was detected by RT-qPCR method, using the primers developed by Lu et al. (Table 4) [27]. RT-qPCR was performed using SsoAdvanced PreAmp Supermix (Bio-Rad Laboratories, Hercules, CA, USA) in a Bio-Rad CFX96 Touch PCR Detection System with the following conditions: 95 °C for 10 min and then 45 cycles of 95 °C for 15 s and 58–66 °C for 30 s, followed by a melt cycle of 65 °C for 5 s and 95 °C for 15 s [39,40]. Reactions were performed in triplicate, with a negative nuclease-free water control in each run. Sweet potato H2 B and UBI encoding genes were used as a double internal control for normalization of the gene expression data [41]. The relative expression levels of virus were quantified with the delta threshold cycle (ΔCt) method [42], referenced to the internal control. The experiments were repeated three times in independent RT-qPCR reactions.

### 4.3. Statistical Analysis

ANOVA tests were preformed using DPS 7.05 software and multiple comparison tests were preformed using Duncan’s new complex difference method (*p* < 0.05), and Rstudio was used for cluster analysis.

## 5. Conclusions

The survival rate of sweet potato plant grafting by insertion method was significantly higher than that by side grafting method. Eighty-six sweet potato varieties were infected with virus by insertion method, and the results of SPVD resistance identification were obtained. The symptomatic test results were verified by NCM-ELISA and qRT-PCR. Among the tested 86 sweet potato varieties, three varieties, Ningzishu No.1, Ning 4-6, and Yushuwang, were identified as susceptible varieties, and three varieties, Wanzi 51, Chaoshu No.1 and Enshu No.2, were identified as highly resistant varieties. In this study, an accurate detection method was established, and could be used for SPVD-resistant sweet potato varieties selection in the production.

## Figures and Tables

**Figure 1 plants-12-00957-f001:**
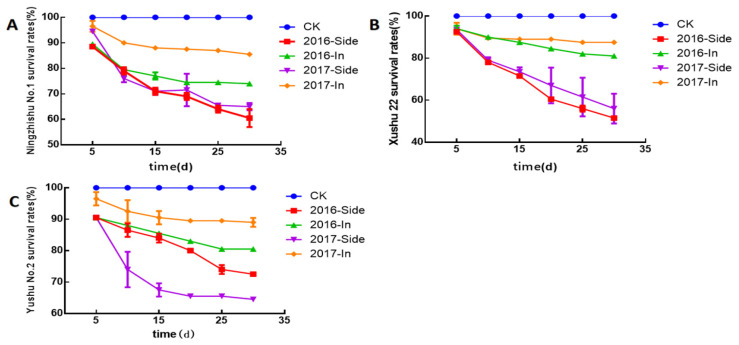
The survival rate of three sweet potato varieties, Ningzhishu No.1 (**A**), Xushu 22 (**B**) and Yushu No.2 (**C**), tested by insertion and side grafting method in years 2016 and 2017. Error bars indicate the standard deviation from two independent replicates.

**Figure 2 plants-12-00957-f002:**
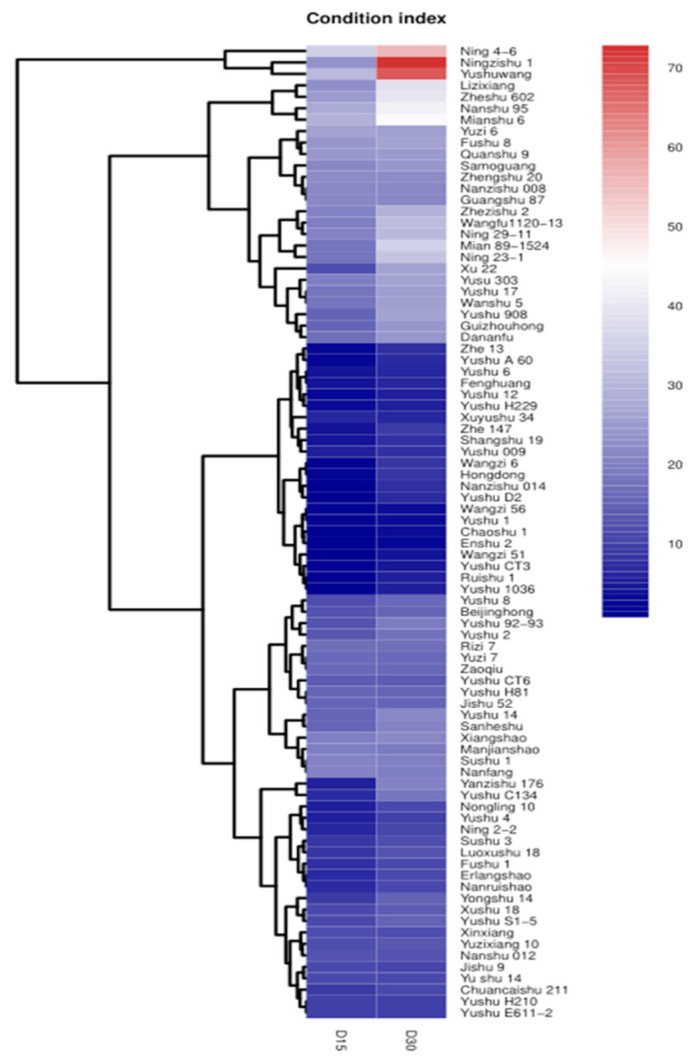
Cluster analysis was performed in terms of disease indexes of 86 sweet potato varieties after graft inoculation of SPVD using the normalized log2-transformed RPKM values of disease index. Red color represents the disease index between 65 and 70, which were identified as susceptible varieties. The blue color indicates that the disease index is between 0 and 27, and those were identified as resistant varieties. The scale bar represents reads per kilobase of exon model per million mapped reads (RPKM) values.

**Figure 3 plants-12-00957-f003:**
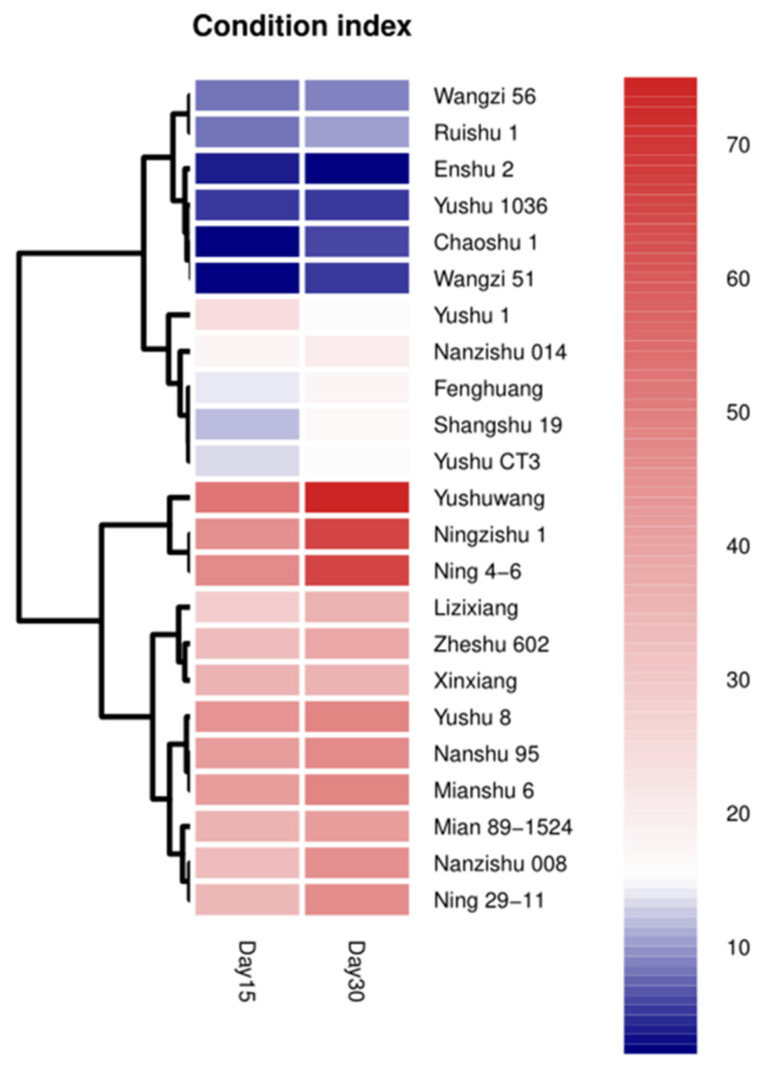
Cluster analysis was performed to group disease index of 23 sweet potato varieties after graft inoculation of SPVD based on the normalized log2-transformed RPKM values of disease index. Red colour represents the disease index between 28 and 70 for susceptible varieties. The blue color indicates that the disease index is between 0 and 27 for resistant variety. The scale bar represents reads per kilobase of exon model per million mapped reads (RPKM) values.

**Figure 4 plants-12-00957-f004:**
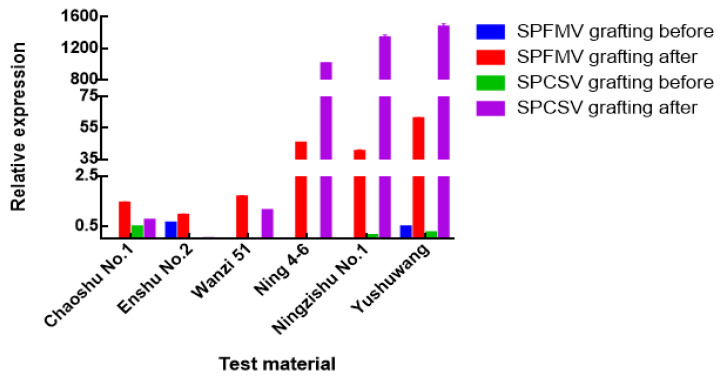
QRT-PCR was used to detect the relative expression levels of genes of SPCSV and SPFMV in 2017. Six varieties, Ningzishu No.1, Ning 4-6, Yushuwang, Wanzi 51, Chaoshu No.1 and Enshu No.2, were sampled before and after grafting with virus seedlings. Error bars indicate the standard deviation from three independent replicates.

**Figure 5 plants-12-00957-f005:**
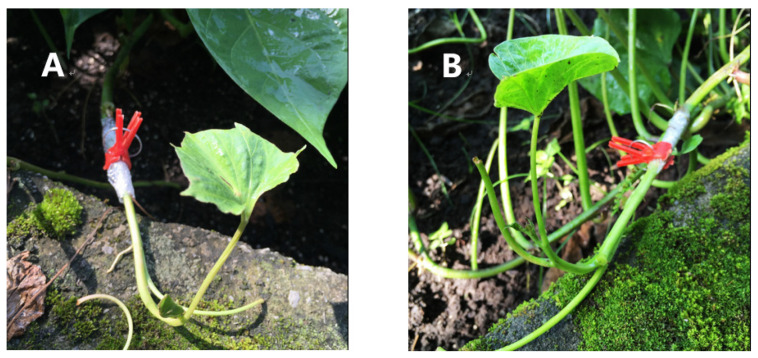
Insertion grafting and side grafting methods. (**A**) The effect of the insertion grafting method after grafting. (**B**) The effect of the side grafting method after grafting.

**Table 1 plants-12-00957-t001:** Disease indexes of 86 sweet potato varieties after inoculation of sweet potato virus disease (SPVD) using grafting in 2016.

Variety Name	Origin	Disease Index Calculated at 15 DAG	Disease Index Calculated at 30 DAG
Enshu No.2	Hubei, China	1.43 ± 0 uvTU	2.86 ± 2.02 yT
Wanzi 56	Chongqing, China	1.43 ± 2.02 uvTU	3.57 ± 3.03 xyST
Yushu No.1	Chongqing, China	2.14 ± 1.01 tuvTU	3.57 ± 3.03 xyST
Chaoshu No.1	Fujian, China	1.43 ± 0 uvTU	3.57 ± 3.03 xyST
Wanzi 51	Chongqing, China	0.71 ± 1.01 vU	4.29 ± 0.00 wxyRST
Yushu CT3	Chongqing, China	1.43 ± 0 uvTU	5.00 ± 1.01 vwxyQRST
Ruishu No.1	Gansu, China	1.43 ± 0 uvTU	5.71 ± 0.00 uvwxyPQRST
Yushu 12	Chongqing, China	2.86 ± 0 stuvSTU	5.71 ± 0.00 uvwxyPQRST
Yushu H229	Chongqing, China	3.57 ± 3.03 rstuvRSTU	5.71 ± 4.04 uvwxyPQRST
Xuyushu 34	Chongqing, China	6.43 ± 1.01 pqrstuvPQRSTU	6.43 ± 1.01 uvwxyOPQRST
Yushu 1036	Chongqing, China	1.43 ± 2.02 uvTU	6.43 ± 1.01 uvwxyOPQRST
Yushu A 60	Chongqing, China	2.86 ± 0 stuvSTU	7.14 ± 0.00 tuvwxyOPQRST
Yushu D2	Chongqing, China	1.43 ± 0 uvTU	7.14 ± 2.02 tuvwxyOPQRST
fenghuang	Chongqing, China	4.29 ± 2.02 rstuvQRSTU	7.14 ± 2.02 tuvwxyOPQRST
Shangshu 19	Henan, China	4.29 ± 2.02 rstuvQRSTU	7.86 ± 1.01 stuvwxyNOPQRST
Nanzishu 014	Sichuan, China	0.71 ± 1.01 vU	7.86 ± 1.01 stuvwxyNOPQRST
Yushu 009	Chongqing, China	5.71 ± 2.02 qrstuvPQRSTU	7.86 ± 1.01 stuvwxyNOPQRST
Yushu No.6	Chongqing, China	2.86 ± 0.00 stuvSTU	7.86 ± 3.03 stuvwxyNOPQRST
Wanzi No.6	Chongqing, China	2.14 ± 1.01 tuvTU	8.57 ± 2.02 rstuvwxyMNOPQRST
Hongdong	Japan	2.14 ± 1.01 tuvTU	8.57 ± 0.00 rstuvwxyMNOPQRST
Zhe 13	Zhejiang, China	2.86 ± 0.00 stuvSTU	8.57 ± 2.02 rstuvwxyMNOPQRST
Zhe 147	Zhejiang, China	4.29 ± 0.00 rstuvQRSTU	9.29 ± 5.05 qrstuvwxyMNOPQRST
Yushu No.4	Chongqing, China	5.71 ± 0.00 qrstuvPQRSTU	10.00 ± 4.04 pqrstuvwxyLMNOPQRST
Yushu E611-2	Chongqing, China	10.00 ± 0.00 mnopqrLMNOPQRST	10.00 ± 0.00 pqrstuvwxyLMNOPQRST
Yushu H210	Chongqing, China	10.00 ± 2.02 mnopqrLMNOPQRST	10 ± 2.02 pqrstuvwxyLMNOPQRST
Yushu 14	Henan, China	11.43 ± 2.02 lmnopqKLMNOPQRS	11.43 ± 2.02 opqrstuvwxyKLMNOPQRST
Ning 2-2	Ningxia, China	6.43 ± 5.05 pqrstuvPQRSTU	11.43 ± 0.00 opqrstuvwxyKLMNOPQRST
Erlangshao	Chongqing, China	7.14 ± 2.02 pqrstuvOPQRSTU	11.43 ± 2.02 opqrstuvwxyKLMNOPQRST
Nongling 10	Chongqing, China	5.71 ± 0.00 qrstuvPQRSTU	11.43 ± 2.02 opqrstuvwxyKLMNOPQRST
Fushu No.1	Fujian, China	7.86 ± 1.01 opqrstuNOPQRSTU	11.43 ± 6.06 opqrstuvwxyKLMNOPQRST
Nanruishao	Yunnan, China	7.14 ± 0.00 pqrstuvOPQRSTU	11.43 ± 0.00 opqrstuvwxyKLMNOPQRST
Xinxiang	Zhejiang, China	20.00 ± 2.02 efghijDEFGHIJK	12.14 ± 3.03 nopqrstuvwxyKLMNOPQRST
Sushu No.3	Suzhou, China	8.57 ± 0.00 opqrstMNOPQRSTU	12.14 ± 5.05 nopqrstuvwxyKLMNOPQRST
Chuancaishu 211	Sichuan, China	9.29 ± 1.01 nopqrsMNOPQRSTU	12.14 ± 1.01 nopqrstuvwxyKLMNOPQRST
Nanshu 012	Sichuan, China	12.86 ± 0.00 klmnopIJKLMNOPQ	12.86 ± 0.00 mnopqrstuvwxyJKLMNOPQRST
Luoxushu 18	Jiangsu, China	8.57 ± 0.00 opqrstMNOPQRSTU	12.86 ± 2.02 mnopqrstuvwxyJKLMNOPQRST
Jishu No.9	Shandong, China	11.43 ± 0.00 lmnopqKLMNOPQRS	13.57 ± 5.05 mnopqrstuvwxyJKLMNOPQRST
Yuzixiang 10	Chongqing, China	12.14 ± 7.07 klmnopqJKLMNOPQR	13.57 ± 1.01 mnopqrstuvwxyJKLMNOPQRST
Yushu CT6	Chongqing, China	14.29 ± 0.00 jklmnoHIJKLMNOP	14.29 ± 0.00 lmnopqrstuvwxyIJKLMNOPQRST
Xushu 18	Jiangsu, China	11.43 ± 0.00 lmnopqKLMNOPQRS	15.00 ± 1.01 klmnopqrstuvwxyHIJKLMNOPQRST
Yushu S1-5	Chongqing, China	11.43 ± 0.00 lmnopqKLMNOPQRS	15.71 ± 2.02 jklmnopqrstuvwxHIJKLMNOPQRST
Yushu H81	Chongqing, China	15.71 ± 0.00 ijklmnGHIJKLMNO	15.71 ± 0.00 jklmnopqrstuvwxHIJKLMNOPQRST
Yongshu 14	Chongqing, China	9.29 ± 3.03 nopqrsMNOPQRSTU	15.71 ± 10.1 jklmnopqrstuvwxHIJKLMNOPQRST
Jishu 52	Hebei, China	15.71 ± 6.06 ijklmnGHIJKLMNO	15.71 ± 6.06 jklmnopqrstuvwxHIJKLMNOPQRST
Zaoqiu	Hebei, China	16.43 ± 9.09 hijklmFGHIJKLMN	16.43 ± 9.09 jklmnopqrstuvwHIJKLMNOPQRST
Yushu No.8	Chongqing, China	8.57 ± 0.00 opqrstMNOPQRSTU	16.43 ± 3.03 jklmnopqrstuvwHIJKLMNOPQRST
Yuzi No.7	Chongqing, China	16.43 ± 1.01 hijklmFGHIJKLMN	16.43 ± 1.01 jklmnopqrstuvwHIJKLMNOPQRST
Beijinghong	Beijing, China	12.86 ± 2.02 klmnopIJKLMNOPQ	16.43 ± 3.03 jklmnopqrstuvwHIJKLMNOPQRST
Rizi No.7	Japan	17.14 ± 6.06 ghijklFGHIJKLM	17.14 ± 6.06 ijklmnopqrstuvGHIJKLMNOPQRST
Yushu No.2	Chongqing, China	12.14 ± 1.01 klmnopqJKLMNOPQR	17.86 ± 1.01 ijklmnopqrstuGHIJKLMNOPQRST
Yushu C134	Chongqing, China	7.14 ± 0.00 pqrstuvOPQRSTU	19.29 ± 15.15 hijklmnopqrstGHIJKLMNOPQRST
Manjianshao	Chongqing, China	20.00 ± 0.00 efghijDEFGHIJK	19.29 ± 1.01 hijklmnopqrstGHIJKLMNOPQRST
Yushu 92-93	Chongqing, China	12.86 ± 0.00 klmnopIJKLMNOPQ	20.00 ± 4.04 hijklmnopqrsFGHIJKLMNOPQRS
Sanheshu	Sichuan, China	15.71 ± 0.00 ijklmnGHIJKLMNO	20.71 ± 3.03 hijklmnopqrFGHIJKLMNOPQR
Yanzishu 176	Shandong, China	5.71 ± 0.00 qrstuvPQRSTU	20.71 ± 5.05 hijklmnopqrFGHIJKLMNOPQR
Yushu 14	Chongqing, China	15.71 ± 4.04 ijklmnGHIJKLMNO	21.43 ± 6.06 ghijklmnopqFGHIJKLMNOPQ
Nanzishu 008	Chongqing, China	21.43 ± 6.06 defghiCDEFGHI	21.43 ± 6.06 ghijklmnopqFGHIJKLMNOPQ
Guangshu 87	Guangdong, China	21.43 ± 2.02 defghiCDEFGHI	21.43 ± 2.02 ghijklmnopqFGHIJKLMNOPQ
Xiangshao	Sichuan, China	20.00 ± 0.00 efghijDEFGHIJK	21.43 ± 8.08 ghijklmnopqFGHIJKLMNOPQ
Zhengshu 20	Zhengzhou, China	22.14 ± 3.03 defghiCDEFGH	22.14 ± 3.03 ghijklmnopFGHIJKLMNOP
Nanfang	Chongqing, China	20.71 ± 1.01 efghijCDEFGHIJ	22.86 ± 2.02 ghijklmnoEFGHIJKLMNO
Sushu No.1	Jiangsu, China	20.71 ± 1.01 efghijCDEFGHIJ	22.86 ± 2.02 ghijklmnoEFGHIJKLMNO
Guizhouhong	Guizhou, China	15.71 ± 0.00 ijklmnGHIJKLMNO	24.29 ± 2.02 fghijklmnDEFGHIJKLMN
Samoguang	Japan	21.43 ± 0.00 defghiCDEFGHI	24.29 ± 0 fghijklmnDEFGHIJKLMN
Dananfu	Sichuan, China	17.14 ± 0.00 ghijklFGHIJKLM	25.00 ± 1.01 fghijklmDEFGHIJKLM
Quanshu No.9	Fujiang, China	24.29 ± 0.00 cdefBCDEFG	25.00 ± 3.03 fghijklmDEFGHIJKLM
Yushu 908	Chongqing, China	15.71 ± 4.04 ijklmnGHIJKLMNO	26.43 ± 1.01 efghijklDEFGHIJKL
Yushu 8	Henan, China	25.00 ± 1.01 bcdefBCDEF	26.43 ± 1.01 efghijklDEFGHIJKL
Yushu 17	Chongqing, China	18.57 ± 0.00 fghijkEFGHIJKL	26.43 ± 3.03 efghijklDEFGHIJKL
Fushu No.8	Fujian, China	24.29 ± 0.00 cdefBCDEFG	26.43 ± 5.05 efghijklDEFGHIJKL
Xu 22	Jiangsu, China	12.14 ± 1.01 klmnopqJKLMNOPQR	26.43 ± 3.03 efghijklDEFGHIJKL
Yusu 303	Chongqing, China	20.00 ± 0.00 efghijDEFGHIJK	27.14 ± 0.00 efghijkDEFGHIJK
Wanshu No.5	Chongqing, China	18.57 ± 4.04 fghijkEFGHIJKL	27.86 ± 11.11 efghijDEFGHIJK
Yuzi No.6	Chongqing, China	26.43 ± 1.01 bcdeBCDE	27.86 ± 1.01 efghijDEFGHIJK
Zhezishu No.2	Zhejiang, China	20.71 ± 1.01 efghijCDEFGHIJ	29.29 ± 1.01 defghiDEFGHIJ
Ning 29-11	Jiangsu, China	20.71 ± 5.05 efghijCDEFGHIJ	30.71 ± 7.07 defghDEFGHI
Wanfu 1120-13	Zhejiang, China	12.14 ± 3.03 klmnopqJKLMNOPQR	31.43 ± 6.06 defghDEFGH
Ning 23-1	Jiangsu, China	18.57 ± 4.04 fghijkEFGHIJKL	33.57 ± 3.03 defgCDEFG
Mian 89-1524	Sichuan, China	18.57 ± 10.1 fghijkEFGHIJKL	35.71 ± 14.14 defBCDEF
Lizixiang	Chongqing, China	22.86 ± 8.08 cdefghBCDEFGH	38.57 ± 14.14 cdeBCDE
Zheshu 602	Zhejiang, China	25.00 ± 5.05 bcdefBCDEF	40.00 ± 6.06 cdBCD
Nanshu 95	Sichuan, China	27.86 ± 1.01 bcdABCD	47.86 ± 1.01 cBC
Mianshu No.6	Sichuan, China	29.29 ± 3.03 abcABC	48.57 ± 10.1 cB
Ning 4-6	Jiangsu, China	35.00 ± 1.01 aA	63.57 ± 11.11 bA
Yushuwang	Henan, China	30.71 ± 1.01 abAB	72.14 ± 11.11 abA
Ningzishu No.1	Jiangsu, China	23.57 ± 1.01 cdefgBCDEFG	75.00 ± 13.13 aA

Data show mean ± SD. Different uppercase and lowercase letters in the same column indicate significant difference at *p* < 0.01 and *p* < 0.05 levels by Duncan’s test, respectively. DAG, days after grafting.

**Table 2 plants-12-00957-t002:** Disease indexes of 23 varieties after graft inoculation of SPVD in 2017.

Variety Name	15-Day Disease Index	Variety Name	30-Day Disease Index
Wanzi 51	2.14 ± 1.01 iH	Enshu No.2	2.86 ± 0 ijF
Enshu No.2	2.14 ± 1.01 iH	Chaoshu No.1	5.00 ± 1.01 ijEF
Chaoshu No.1	2.14 ± 1.01 iH	Wanzi 51	5.00 ± 1.01 ijEF
Yushu No.1	5.00 ± 1.01 hiH	Yushu 1036	5.71 ± 0.00 ijEF
Yushu 1036	5.00 ± 5.05 hiH	Yushu No.1	6.43 ± 3.03 ijEF
Wanzi 56	7.86 ± 1.01 ghiGH	Wanzi 56	9.29 ± 1.01 hiDEF
Ruishu No.1	8.57 ± 2.02 ghiGH	Ruishu No.1	10.71 ± 3.03 ghiDEF
Shangshu 19	12.14 ± 1.01 fghGH	Yushu CT3	15.71 ± 2.02 fghDE
Yushu CT3	13.57 ± 1.01 fghGH	Shangshu 19	17.14 ± 2.02 fgD
Fenghuang	14.29 ± 4.04 fgGH	Fenghuang	17.86 ± 1.01 fgD
Nanzishu 014	17.86 ± 3.03 fFG	Nanzishu 014	20.00 ± 2.02 fD
Lizixiang	28.57 ± 0.00 eEF	Lizixiang	35.71 ± 2.02 eC
Zheshu 602	32.86 ± 2.02 deDE	Yushu No.8	35.71 ± 2.02 eC
Nanzishu 008	32.86 ± 2.02 deDE	Zheshu 602	38.57 ± 6.06 deBC
Ning 29-11	34.29 ± 8.08 deCDE	Mian 89-1524	41.43 ± 6.06 cdeBC
Yushu No.8	35.71 ± 10.1 cdeBCDE	Nanzishu 008	45.71 ± 8.08 cdBC
Mian 89-1524	35.71 ± 2.02 cdeBCDE	Ning 29-11	46.43 ± 1.01 cBC
Nanshu 95	41.43 ± 2.02 bcdABCD	Nanshu 95	47.14 ± 6.06 cB
Mianshu No.6	41.43 ± 2.02 bcdABCD	Xinxiang	48.57 ± 4.04 cB
Xinxiang	44.29 ± 2.02 abcABCD	Mianshu No.6	48.57 ± 0.00 cB
Ningzishu No.1	45.71 ± 4.04 abABC	Ning 4-6	65.71 ± 4.04 bA
Ning 4-6	47.14 ± 6.06 abAB	Ningzishu No.1	65.71 ± 4.04 bA
Yushuwang	52.86 ± 6.06 aA	Yushuwang	75.00 ± 1.01 aA

Data show mean ± SD. Different uppercase and lowercase letters in the same column indicate significant difference at *p* < 0.01 and *p* < 0.05 levels by Duncan’s test, respectively. DAG, days after grafting.

**Table 3 plants-12-00957-t003:** NCM-ELISA Detection Results of 86 sweet potato varieties after graft inoculation of SPVD in 2016.

Variety Name	NCM-ELISA		Variety Name	NCM-ELISA	
	SPCSV	SPFMV		SPCSV	SPFMV
Enshu No.2	−	−	Jishu 52	−	−
Wanzi 56	−	−	Zaoqiu	−	+
Yushu No.1	−	−	Yushu No.8	−	−
Chaoshu No.1	−	−	Yuzi No.7	−	−
Wanzi 51	−	−	Beijinghong	−	−
Yushu CT3	−	−	Rizi No.7	−	−
Ruishu No.1	−	−	Yushu No.2	−	−
Yushu 12	−	−	Yushu C134	−	−
Yushu H229	−	−	Manjianshao	−	−
Xuyushu 34	−	−	Yushu 92-93	−	−
Yushu 1036	−	−	Sanheshu	−	+
Yushu A 60	−	−	Yanzishu 176	−	+
Yushu D2	−	−	Yushu 14	−	−
Fenghuang	−	−	Nanzishu 008	+	+
Shangshu 19	−	−	Guangshu 87	−	+
Nanzishu 014	−	−	Xiangshao	−	+
Yushu 009	−	−	Zhengshu 20	−	+
Yushu No.6	−	−	Nanfang	−	+
Wanzi No.6	−	−	Sushu No.1	−	+
Hongdong	−	−	Guizhouhong	−	+
Zhe 13	−	−	Samoguang	−	−
Zhe 147	−	−	Dananfu	−	+
Yushu No.4	−	−	Quanshu No.9	−	+
Yushu E611-2	−	−	Yushu 908	−	+
Yushu H210	−	−	Yushu 8	−	−
Yushu 14	−	+	Yushu 17	−	+
Ning 2-2	−	−	Fushu No.8	−	+
Erlangshao	−	−	Xu 22	−	+
Nongling 10	−	−	Yusu 303	−	+
Fushu No.1	−	−	Wanshu No.5	−	+
Nanruishao	−	−	Yuzi No.6	−	+
Xinxiang	+	+	Zhezishu No.2	−	+
Sushu No.3	−	−	Ning 29-11	+	+
Chuancaishu 211	−	−	Wanfu 1120-13	−	−
Nanshu 012	−	−	Ning 23-1	−	+
Luoxushu 18	−	−	Mian 89-1524	+	+
Jishu No.9	−	−	Lizixiang	+	+
Yuzixiang 10	−	−	Zheshu 602	+	+
Yushu CT6	−	−	Nanshu 95	+	+
Xushu 18	−	−	Mianshu No.6	+	+
Yushu S1-5	−	−	Ning 4-6	+	+
Yushu H81	−	−	Yushuwang	+	+
Yongshu 14	−	−	Ningzishu No.1	+	+

“+” indicates positive reaction with SPFMV or SPCSV polyclonal antibody. “−” represents a negative reaction with SPFMV or SPCSV polyclonal antibodies.

**Table 4 plants-12-00957-t004:** Specific primers for RT-qPCR detection.

Primer Name	Forward Reverse Primer	Size of PCR Product (bp)
F (SPFMV)	5′-TGTGCCTCTCCGTATCYTCTTCTTGCGT-3′	149
R (SPFMV)	5′-GACTGATATGAGTCTTGCGCGRTATGCG-3′	
F (SPCSV)	5′-CCCAACGTGTTTATCTATTACTAAGAGTGG-3′	170
R (SPCSV)	5′-AATACTGGGGAGCTATCTTACGTTTGA-3′	
RIbUBI	5′-CTTGAT CTTCTTCGGCTTGG-3′	
FIbH2 B	5′-GTGCCGGAGACAAGAAGAAG- 3′	110
RIbH2 B	5′-CTTGCTGGAGATTCCGATGT-3′	

## Data Availability

The datasets generated during and/or analyzed during the current study are available from the corresponding author on reasonable request.

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
