# Peer review of "Application of Grafting Method in Resistance Identification of Sweet Potato Virus Disease and Resistance Evaluation of Elite Sweet Potato [Ipomoea batatas (L.) Lam] Varieties"

_plants, 2023, doi:10.3390/plants12040957_

Round 1

Reviewer 1 Report

Table 3

I recommend you to arrange the order of the variety names for the ELISA detection according to the order of the disease index (DI) results in 30DAG in Table 1. This makes it easier to compare the DI with the SPVD infection status. It also makes it easier to recognize the possibility of resistant or tolerant varieties.

 Figure 5

The photos is difficult to understand. In photograph A, the rootstock is too dark to be seen. In photograph B, the direction of the rootstock cannot be recognized. I cannot understand which direction is the growing point. Please make it easier to understand.

 4.2.2 Virus inoculation

When assessing viral resistance, I think that the infected scions should completely adhere to the subjected rootstocks. Did the authors confirm that infected scion-rootstock adhesion was established for all varieties?

 4.2.3 Disease index calculation and cluster analysis

In the case of insertion grafting, it is unclear how the rootstocks grew. Did you investigate the symptom expression of each variety grown from axillary bud of rootstocks? The authors need to describe the rootstock growth process in detail.

 That is all.

Author Response

Thanks for your suggestions and please see the attachment.

Reviewer 2 Report

PAY ATTENTION TO THE REFERENCES IN THE WHOLE TEXT: WRITE THE PROPER NUMBERS, NOT THE AUTHORS AND YEARS!

English proofreading is necessary for improving the quality of the manuscript

Line 4 – write the latin name and the author

Line 15 – „Sweet“ – not in bold

Line 15 – write the latin name and author after „sweet potato“

Line 19 – instead of „3“ write „three“

Line 25 instead of „different“, write „tested“

Line 28 – „grafting“ ,„symptomatic“ and „QRT-PCR“ write in lowercase

Line 96 – after „30 days“ write „period“

Line 131 -132; 155-156; 183-184 – write the blue text in black colour without highlighting the text

Line 154 – „Enshu No.2“ write  in black colour

Line – 175 write „there were 12 varieties positive“

Line 176 – write „both by SPCSV“

Line 178, 179, 180, 187, 188 – write „weren't infected by SPVD“ instead of „didn't have SPVD“

Line 187 write „were negative on“

Line 190 – write „Yushu No.8, which was positive“

Line 191 – instead of „of SPFMV“ and „for SPFMV“ write „on SPFMV“

Line 191 – delete „while“

Line 194 – delete „6“ and write „six“

Line 194 – between „grafting“ and „indicating“ delete the space

Line 201-202 -write „methods with the accurancy of the results“

Line 213 – instead of „While“  write “On the other side,“

Line 215 – write “and in conclusion the grafting“

Line 262-263 – write „showed that the accumulation of“

Line 272 – write „plants also through mechanical“

Line 284 – delete the space between „addition,“ and „NCM-ELISA“

Line 291 – „might be that SPVD“

Line 294 „it might be because“

Line 311 – after „Xushu“ write comma „ ,“

Line 316 – „16.0 m“

Line 320 –  after „(Fig. 5, A)“   write „consist of these steps: the single segment…“

Line 325 – after „(Fig. 5, B)“ write „is described by „….(write the proper number of reference for Wang et al., 2014)

Line 326 – write „cut out about 1 cm-length“

Line 330-331 – do you have better quality photos?

Line 352 – put space between „grafting“ and „15“

Line 366 – after „CIP“ - write the city and country

Line 367 – „according to the manufacturer's instructions“

Line 376 – you mean Table 1 or Table 5?

Author Response

(The authors gave the same response as above.)
